# Peer review of "The Clinical Importance of IL-6, IL-8, and TNF-α in Patients with Ovarian Carcinoma and Benign Cystic Lesions"

_diagnostics, 2021, doi:10.3390/diagnostics11091625_

Round 1
Reviewer 1 Report
I have made suggestions and raised questions within the manuscript itself, which I am uploading to be conveyed to the authors. I believe their results are sound and recommend acceptance after making the modifications as follows:
- Introduce a table and presentation on the combined diagnostic parameter (lines 274-275) for IL-8 & TNFalpha in combination using the cutoffs that they have defined.
- Include the data in a table or text for the statement at lines 281-282, comparing Stage I with other stages of OVCA.

Reviewer 2 Report
The present manuscript describes the diagnostic power of some circulating cytokines in distinguishing between benign lesions and adenocarcinoma of ovaries (OC). The authors demonstrate an independent significant correlation between some cytokines and the occurrence of OC. The manuscript has some potential interest in the field; however, some important changes are required in order to reach the publication priority.
- The authors should specify better the histopathological classification of the tumours showing also some representative images.
- The authors should perform an analysis of the results stratified for histological subtypes and FIGO staging and reproduce their data in the different sub-populations.
- The authors should stratify the population for more than two-fold and more than three-fold the upper limit of Ca125 and HE4 and repeat the correlation analysis with the cytokines levels. This should be useful in identifying a subpopulation in which the determination of the cytokines should be required.
- The statistical analysis should be repeated with a conventional software used for this kind of analysis (i.e.: SPSS).
- The authors should also perform an analysis of the correlation between cytokine levels and the survival of the patients affected by OC.
- Some misreading in the text should be carefully corrected.
Round 2
Reviewer 2 Report
The authors have tried to address some concerns and give explanations for the other.